# A Direct Silanization Protocol for Dialdehyde Cellulose

**DOI:** 10.3390/molecules25102458

**Published:** 2020-05-25

**Authors:** Arianna Lucia, Markus Bacher, Hendrikus W. G. van Herwijnen, Thomas Rosenau

**Affiliations:** 1Wood K Plus–Competence Center for Wood Composites and Wood Chemistry, Kompetenzzentrum Holz GmbH, Altenberger Straße 69, A-4040 Linz, Austria; a.lucia@wood-kplus.at (A.L.); e.herwijnen@wood-kplus.at (H.W.G.v.H.); 2Institute for Chemistry of Renewable Resources, University of Natural Resources and Life Science Vienna (BOKU), Konrad-Lorenz-Straße 24, A-3430 Tulln an der Donau, Austria; markus.bacher@boku.ac.at; 3Johan Gadolin Process Chemistry Centre, Åbo Akademi University, Porthansgatan 3, FI-20500 Åbo/Turku, Finland

**Keywords:** biomaterials, cellulose, dialdehyde cellulose, organosilane chemistry, ^29^Si NMR, solid state NMR, silanization

## Abstract

Cellulose derivatives have many potential applications in the field of biomaterials and composites, in addition to several ways of modification leading to them. Silanization in aqueous media is one of the most promising routes to create multipurpose and organic–inorganic hybrid materials. Silanization has been widely used for cellulosic and nano-structured celluloses, but was a problem so far if to be applied to the common cellulose derivative “dialdehyde cellulose” (DAC), i.e., highly periodate-oxidized celluloses. In this work, a straightforward silanization protocol for dialdehyde cellulose is proposed, which can be readily modified with (3-aminopropyl)triethoxysilane. After thermal treatment and freeze-drying, the resulting product showed condensation and cross-linking, which was studied with infrared spectroscopy and ^13^C and ^29^Si solid-state nuclear magnetic resonance (NMR) spectroscopy. The cross-linking involves both links of the hydroxyl group of the oxidized cellulose with the silanol groups (Si-O-C) and imine-type bonds between the amino group and keto functions of the DAC (-HC=N-). The modification was achieved in aqueous medium under mild reaction conditions. Different treatments cause different levels of hydrolysis of the organosilane compound, which resulted in diverse condensed silica networks in the modified dialdehyde cellulose structure.

## 1. Introduction

The booming developments in the biopolymers field is evidently engaging cellulose and cellulose derivatives in crescent number of applications and new materials, such as fillers or matrices in polymer composites, aerogels, and separation media. Cellulosic components are simply central in the evolution of novel bio-materials [1]. One particular modification that is especially promising in the area of organic–inorganic hybrid materials is the silanization of cellulose—or polysaccharides in general—by this way generating polysaccharide–silane/silica interfaces with differing amounts of covalent bonds between the two bordering constituents. Silanization is widespread for enhancing the properties of cellulose composites: the coupling with silane agents involve an improved interfacial adhesion between fibers and matrix, better resistance to water leaching, hydrophobicity, thermostabilization, and improved fiber strength [2,3,4]. Quite many applications of novel cellulosic materials modified by silanes can be found in literature: hybrid substances formed by silica gel and dicarboxylic cellulose for dye absorption [5], films made with cellulose acetate and silane with isocyanate moieties [6]; aerogels with cross-linked cellulose, acrylamide polymers, and methyltrichlorosilane for oil/water separation [7]; or composite aerogels with silica and cellulosic fibers for thermal insulation [8] are just a few examples. The chemistry between cellulose derivatives and modified silica gels is used also in the field of chiral separation, producing packing materials for liquid chromatography [9]. Modifications of cellulosic materials are performed in wet state with different silanes [10,11].

A special cellulose derivative with great reverberation in research and applications is dialdehyde cellulose (DAC), usually produced by oxidation of cellulosics (to different degrees) with sodium periodate. Several applications of this type of oxidized cellulose are reported in the literature, such as films for packaging [12], as nanoparticles for drug delivery systems [13], the formation of nanocrystal aerogels with superabsorbent properties [14], or as self-healing nanocomposite hydrogels [15]. In this work, we describe the direct modification of DAC according to a straightforward silanization protocol, carried out in aqueous media and without severe thermal treatment. The chosen silane is (3-aminopropyl)triethoxysilane (APTES). This reagent is cheap and readily available, which is an important factor when it comes to up-scaling. APTES has been studied already in systems with cellulosic fibers [16]. Its utilization has already been reported for strengthening the interfaces of hybrid organic-inorganic coatings [17], in grafting reaction with tosylcellulose [18], for surface functionalization of cellulose nanocrystals [19], and also with DAC as the stationary phase in chromatography [20]. In the latter case, silica gel was modified by APTES in toluene, and then utilized for the modification of DAC in pyridine at high temperature during several hours which is considered neither practical and general nor compatible with green chemistry principles. In our approach, the silanization of DAC involves hydrolysis of APTES and self-condensation as well as condensation with the hydroxyl groups of the cellulose derivative, most notably using aqueous media and employing mild reaction conditions and short reaction times. The studies leading to the DAC modification protocol are described in the present account.

## 2. Results and Discussion

Periodate oxidation of cellulose, if reaching sufficiently high oxidation degrees of about 60% and above, provides a water-soluble material [21]. During the oxidation and solubilization process a great variety of masked aldehyde structures is formed, mainly aldehyde hydrates, hemiacetals, and hemialdals with intra- and intermolecular bonds [12,22,23]. The determination of the molecular weight of DAC, because of inter-chain crosslinking, is rather complicated and requires special approaches and precautions which have been addressed previously [23] and cannot be discussed here. The degree of oxidation was 59% for the thermally treated DAC and 62% for the freeze-dried DAC in our experiments, which corresponds to 7.4 mmol/g and 7.8 mmol/g of aldehyde groups, respectively. The acidic pH during the periodate oxidation promotes the formation and stability of hemiacetal structures [24], which are detected in the Fourier-transform infrared (FTIR) spectrum of the oxidized cellulose [25] (band at 876 cm^−1^ in Figure 1). Also, the influence of periodate oxidation on cellulose structure and morphology, according to the cellulose allomorphs, has been studied [26]. Since the acidic environment at the same time catalyzes hydrolysis of APTES [27], the organosilane was directly added to the DAC solution without a previous hydrolysis step, which is usually performed when this silane is used with cellulose. It has been reported that cellulose and other cellulose derivatives bind covalently with silanes after thermal treatment [3,18,28,29], although it was by no means clear that the behavior would be similar between DAC and APTES, since the chemistry of DAC—and its reactive moieties—is rather different from that of celluloses. Nevertheless, hydroxyl groups present in DAC in high number because of the solubilization process can be expected to be available for cross-linking and condensation with the silanol groups. A similar reactivity of DAC hydroxyl groups and the hydroxyl groups of polysaccharides with regard to hydrogen bond network formation has already been noted in the literature, e.g., for polyvinyl alcohol [30]. So far, modification of DAC with APTES took only imine formation [20] into account, but not a possible reaction between the hydrolyzed organosilane and DAC (hemiacetal/aldehyde hydrate/hemialdal) hydroxyl groups. While solubilized DAC is a film-like materials after thermal drying and foam-like after lyophilization (freeze-drying), the morphology changes upon derivatization and silanized DAC is a powder.

Covalent bonding between cellulose and organosilane can be studied with ^29^Si NMR: if the spectra do not change anymore over time after an initial period, further self-condensation of the silane structures is blocked by the covalent bonding with the cellulose molecules [28]. In this work we apply this approach to the reaction between APTES and DAC to study presence and type of covalent binding between the cellulose and the silane networks.

Figure 2 shows the cross-polarization/magic angle spinning (CP/MAS) ^29^Si NMR spectra of the DAC-APTES condensation product after thermal treatment, revealing peaks of the two main condensation structures of the silane, namely T^2^ and T^3^ [6], with chemical shifts δ at −60.1 ppm and −68.5 ppm, respectively, in the one-day-old sample and δ −60.3 ppm and −68.4 ppm, respectively, for the same sample after two years. A minor contribution of the T^1^ structure is visible at −50 ppm as a shoulder in both spectra. It was evident that the spectra did not change significantly within the two years of sample storage. Therefore, the silane structure formation is completed already in the first sample and stays constant afterwards, and the silane moieties are almost completely crosslinked in a tridimensional structure after the thermal treatment, involving the covalent bond with DAC. The ^29^Si-CP/MAS NMR spectra did not change over time, because of the bonding between the silanol group and the hydroxyl group in DAC: if those bond had not occurred, hydrolyzed APTES would have continued to be engaged in self-condensation and have formed more highly condensed structure over time, which would have significantly changed the spectra (increase of T^3^).

Given in Scheme 1, the condensation reaction of silanes and DAC occurred between the DAC′s different hydroxyl groups and the silanol groups during the thermal treatment, which withdraws the water, this way shifting the equilibrium. 

Please, note once more that the structures in Scheme 1 are just examples of possible structures for the masked aldehyde groups [23].

In addition to the formation of Si-O-C structures by hydroxyl group condensation, a Schiff-base reaction of (masked) aldehydes with the amino group of APTES can occur. The aldehyde function can be present either in its free form or in its masked forms, which react the same way because of the underlying dynamic equilibria. We started from the hypothesis that the thermal treatment was mainly supporting the condensation of APTES′ amino group with the DAC structures (besides further promoting condensation among the hydrolyzed silanes). The corresponding nucleophilic substitution reactions in the mild media present would initially produce hemiaminal structures, which would need more drastic media for the subsequent water elimination leading to the double bond structure of imines. We thus suspected elevated temperatures to be a suitable means for that, and indeed, imine bond formation was evident under these conditions. Surprisingly, the same happened also when we used the alternative approach of freeze-drying to remove water from the reaction mixture. Also, this much milder approach was obviously sufficient to move the equilibrium toward condensation product formation. Both thermal treatment and freeze-drying were effective enough in removing the reaction water, the second technique being clearly more energy-efficient and much milder. The final product (Scheme 1, bottom), involves both imine moieties as well as the Si-O-C structures among silanes and DAC.

The ^29^Si-CP/MAS NMR of the freeze-dried DAC-APTES in Figure 3 shows the cross-linked structures of the silane networks, but with different proportions with respect to Figure 2. While in Figure 2 (thermally treated sample), T^3^ was most prominent and T^1^ present only in small amounts, Figure 3 (freeze-dried sample) shows a more equal contribution of the three condensed structures with an order of T^2^ > T^1^ > T^3^, meaning that the condensation degree (Si-O-Si) was generally smaller here. In Figure 2, signals of the different silane structures were detected at δ = −49.8, −59.5, −68.6 ppm in the sample after one day, and at δ = −50.1, −59.5, −67.8 ppm in the sample after two months′ storage. The integral ratios of the three peaks did not change over time, showing the stability of the cross-linking even without thermal treatment. This result is different from cellulose reacting with silanization agents [16,28], where the condensation is slowly progressing and changing over much longer times.

The presence of non-reacted APTES, which was concluded from ^29^Si NMR (see caption of Figure 3), can also be seen in the FTIR spectra of the samples (Figure 4). In the freeze-dried sample after one day, three typical bands of neat APTES are visible at 2974 cm^−1^ (methyl group, asymmetrical stretching), at 1388 cm^−1^ (methyl group, deformation), and at 952 cm^−1^ (skeletal vibration), identical with those of pure APTES in Figure 1 [31]. It is known that APTES is moisture-sensitive and degrades nearly completely within 28 days [32]; this explains why the peak intensities decreased and eventually disappeared.

The DAC-APTES structures show the same bands in the FTIR spectra both for thermal and freeze-drying after-treatment. The bands of the O-Si-O network are at 1644 and 1042 cm^−1^: the latter is rather close to one of the C-O bands of DAC at 1016 cm^−1^, so both partly superimpose. In the thermally treated samples a double peak was detected, with the second peak at 1070 cm^−1^, ascribable to siloxane (Si-O-Si) bonds [33]. The strong intensity of this peak can be explained with the higher condensation degree of the silica network, which corresponds well with the results of the ^29^Si NMR. The band at 898 cm^−1^ in the freeze-dried samples is in the range of the free silanol group stretching vibration and partly overlapped with the hemiacetal band in DAC. It shifted to 914 cm^−1^ in case of the thermal treated sample, with the hemiacetal contribution being reduced after thermal treatment. The bands at 762 and 684 cm^−1^ correspond to the C-H deformation in the aliphatic chain and Si-C vibration [34], whereas the band at 1590 cm^−1^ is attributed to the imine bond C=N vibration [35].

In the ^13^C-CP/MAS NMR spectra of the freeze-dried samples, shown in Figure 5A, the trace of non-reacted APTES is clearly visible, the sharp signals at δ = 58.4 ppm and 18.5 ppm (Ca and Cb in the APTES structure of Scheme 1) being identical to those of pure APTES, shown in Figure 5B. Since there were no changes in the condensed structures over time as seen by^29^Si NMR, the decrease of these signals is caused by slow decomposition by atmospheric humidity. The peaks of the aliphatic chain in these APTES hydrolysates/condensates are slightly shifted relative to pure APTES. The hydrolysis—and the corresponding shift of the resonances—started already after about one hour of the reaction [16], and arrived at the ultimate shift values of δ 10.6–10.3 ppm (Cα) and at δ 21.5–21.2 (Cβ) ppm after one day (not changes up to two months). The Cγ peak is broadened and resulted in a shoulder at around 42 ppm. The peaks at around 15.2 ppm in the freeze-dried sample after one day and 14.9 ppm in the freeze-dried sample after two months storage arise from the APTES side chains in condensed oligomers. The ^13^C resonances for the DAC carbons, except C2/C3, are still present in the spectra, with chemical shifts almost identical to non-modified DAC, shown in Figure 5C. Only the small peak of free (non-masked) aldehyde carbons C2/C3 (δ = 201.2 ppm) was not detected any longer, whereas a new signal at δ 170 ppm, characteristic for imine carbons, appeared [35]. The peak corresponding to the carbon in the C-O-Si bridge is located in the region between 60 and 50 ppm, but the overlap of resonances made an unambiguous assignment impossible.

The same characteristic signals and chemical shifts as for the freeze-dried samples can be seen in the spectrum of the thermally treated sample (Figure 6). The resonances from residual non-reacted APTES are absent, because it is fully condensed into the silanes network and with DAC. Furthermore, there is a decrease of the C1 and C4 intensities from DAC demonstrating a more pronounced condensation, and a significant increase in imine-type structures which point to the same conclusion. For both treatment options, thermally induced and freeze-drying, the NMR analyses confirmed the direct silanization of DAC with presence of both DAC-silanol Si-O-C interlinks and imine bonds. Differences in the degree of condensation arise from the different processes of drying, with the severity being higher in the case of the thermal option.

## 3. Materials and Methods

Microcrystalline cellulose (Avicel PH101), sodium metaperiodate (ACS reagent, ≥ 99.8%), acetic acid (glacial, ≥ 99%), and (3-aminopropyl)triethoxysilane (APTES, 99%) were purchased from Sigma-Aldrich (Schnelldorf, Germany).

### 3.1. Cellulose Oxidation and Solubilization

Microcrystalline cellulose was suspended in an aqueous solution (deionized water) of sodium metaperiodate with a 1.25 molar ratio between the oxidant and cellulose (anhydroglucose unit). The suspension was stirred for 24 h in the dark at 35 °C. The product was separated by centrifugation (5000 rpm for 20 min, Hettich Rotina 380, Westphalia, Germany) and washed. The never-dried DAC was suspended in water (5% of solid content) and heated to 100 °C for 90 min for solubilization [36]. The pH was adjusted to 3.5 with acetic acid. Determination of the aldehyde content was performed with the oxime titration method, as reported in the literature [37].

### 3.2. Silanization Protocol

An aliquot of APTES was added to the DAC solution under stirring at 250 rpm at r.t., with a molar ratio of aldehyde: organosilane of 2.5. After 1 h, the precipitated product was collected and washed by centrifugation (5000 rpm for 15 min, 700 mL of water in total per sample). For the freeze-drying treatment, the sample was frozen at −80 °C and then lyophilized (Christ Beta 1–8 LD Plus, Martin Christ Gefriertrocknungsanlagen GmbH, Osterode am Harz, Germany). For the thermal treatment, the sample was placed in an oven at 105 °C for 1 h (Memmert UNB 400, Schwabach, Germany). The samples were stored in vials in a ventilated cupboard.

### 3.3. Solid-State NMR and FTIR Measurements

All solid state NMR experiments were performed on a Bruker Avance III HD 400 spectrometer (Rheinstetten, Germany), resonance frequency of ^1^H at 400.13 MHz, ^13^C at 100.61 MHz, and ^29^Si at 79.54 MHz, respectively, equipped with a 4 mm dual broadband CP/MAS probe. ^13^C spectra were acquired by using the total sideband suppression (TOSS) sequence at ambient temperature with a spinning rate of 5 kHz, a cross-polarization (CP) contact time of 2 ms, a recycle delay of 2 s, SPINAL−64 ^1^H decoupling and an acquisition time of 49 ms whereas the spectral width was set to 250 ppm. ^13^C chemical shifts were referenced externally against the carbonyl signal of glycine at δ = 176.03 ppm. ^29^Si NMR spectra were acquired with the normal CP pulse sequence using a spectral width of 300 ppm and a contact time of 2 ms. Chemical shifts were referenced externally against DSS with δ = 0 ppm.

The samples were analyzed with a PerkinElmer Frontier FTIR Single-Range spectrometer in ATR mode (PerkinElmer Frontier, Waltham, MA, United States)**.**

## 4. Conclusions

In this work we presented a straightforward protocol, which meets common green chemistry principles, for the direct silanization of DAC, one of the most recently developed and studied cellulose derivatives. We describe the direct silanization of DAC with APTES, through thermal treatment and freeze-drying, both with imine formation and hydroxyl group condensation. The grafting and covalent binding between the organic and inorganic counterparts was proven and the structures characterized by ^29^Si solid-state nuclear magnetic resonance, together with infrared spectroscopy (Fourier-transform Infrared, FTIR). Interestingly, condensation occurred not only after thermal treatment (as it does in the case of cellulose), but also after a simple freeze-drying process. The condensation of the hemiacetal/aldehyde hydrate/hemialdal groups from DAC and the silanol groups in the hydrolyzed APTES occurred simultaneously with imine link formation between the masked aldehyde structures of DAC in APTES′ amino group. The resulting material was characterized by NMR techniques: the absence of changes in the spectra over time confirmed the proposed cross-linking and its stability. The constancy of the spectra indicated that formation of the crosslinked network was completed after one day and did not proceed further or change afterwards. Apparently, the reaction centers were consumed or became inaccessible because of the decreased internal mobility of the structure. In any case, this relatively fast process in the DAC case is different from the slow, continuously changing process in the case of celluloses as the co-reactant of silanes.

The modification was achieved in aqueous media and with mild reaction conditions, even by an energy-saving and byproduct-reducing freeze-drying process instead of a thermal treatment. ^13^C CP/MAS NMR confirms different grades of hydrolysis and condensation severity, depending on the drying process. In all cases, the imine bond is confirmed. The product showed a larger silane network for the specimens after thermal treatment. FTIR spectra confirmed all conclusions derived from NMR. The resulting condensed hybrid product, combining an organic and inorganic phase, represent a class of important biomaterials with diverse applications reaching from materials science over separation science and chromatography to medicine.

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
