# Peer review of "A Direct Silanization Protocol for Dialdehyde Cellulose"

_molecules, 2020, doi:10.3390/molecules25102458_

Round 1

Reviewer 1 Report

In this manuscript, authors present a protocol for direct silanization of DAC. The introduction is concise, but provides a good overview of the state of the art and potential applications of modified DAC in several fields. Methods are well described and conclusions are supported by the results. 

The manuscript can be accepted in its present form.

Author Response

No changes required.

Reviewer 2 Report

The authors describe a protocol to perform the silanization of a cellulose derivative, dialdehyde cellulose. NMR and FTIR were used to characterize the samples.

The following minor points can be addressed:

The molecular weight of the samples, initial and final, should be determined.  The protocol implies the use of mild reaction conditions however if materials science applications are target (as suggested by authors at the end of the conclusions) the molecular weight of the samples is crucial for the mechanical properties of the materials produced.

The authors speak about a network formation. Can the distance between crosslinks be determined? In Scheme I, first reaction, shouldn´t cellobiose be consider?

A more detailed explanation should be given to the following phrase: …”The formation of the crosslinked network was completed after one day and did not proceed further or change…” Why? No more available sites to crosslink reaction? Does the crosslinked network depend on the average molecular weight of cellulose?

Author Response

The authors describe a protocol to perform the silanization of a cellulose derivative, dialdehyde cellulose. NMR and FTIR were used to characterize the samples.
The following minor points can be addressed:
The molecular weight of the samples, initial and final, should be determined. The protocol implies the use of mild reaction conditions however if materials science applications are target (as suggested by authors at the end of the conclusions) the molecular weight of the samples is crucial for the mechanical properties of the materials produced.

We completely agree with this statement. The molecular weight is a crucial parameter, also for DAC. The determination is especially complicated due to the dynamic equilibria of hemiacetals/ hemialdals and require special measurement conditions. This has been addressed in a separate paper (Sulaeva et al.) which has been cited. This complex topic exceeds the scope of the paper by far.
We have added the following sentence in the first paragraph of the results section:
"The determination of the molecular weight of DAC, because of inter-chain crosslinking, is rather complicated and requires special approaches and precautions which have been addressed previously [23] and cannot be discussed here."

The authors speak about a network formation. Can the distance between crosslinks be determined?

Unfortunately, this is impossible due to the heterogeneity (with regard to distribution and chemistry) of the structures.

In Scheme I, first reaction, shouldn´t cellobiose be consider?

Cellobiose is the repeating unit in the solid state structure, glucose (glucopyranose) in the chemical structure. In addition, the duplication of the unit in the formula scheme would not provide any additional information, but would only make it more complicated for the reader, so we would prefer to leave the scheme as it is.

A more detailed explanation should be given to the following phrase: …”The formation of the crosslinked network was completed after one day and did not proceed further or change…” Why?
No more available sites to crosslink reaction? Does the crosslinked network depend on the average molecular weight of cellulose?

The kinetics of the crosslinking reaction are faster in the DAC case compared to cellulose. Giving reasons for that would be largely speculative. To make the difference between DAC and cellulose more clear, we changed the text as follows (first paragraph of conclusion section):
"The constancy of the spectra indicated that formation of the crosslinked network was completed after one day and did not proceed further or change afterwards. Apparently the reaction centers were consumed or became inaccessible due to the decreased internal mobility of the structure. In any case, this relatively fast process in the DAC case is…"

Reviewer 3 Report

This manuscript reports the straightforward silanization protocol for the preparation of dialdehyde cellulose hybrid composites. The experiment and their results were well discussed throughout the manuscript. However, there are lacks of the physicochemical properties related to potential applications, therefore which does not give the higher impact and also novelty on these research categories. For instance, the mechanical properties and surface nature (hydrophobic and hydrophilic) should be also kindly provided at least. In addition, importantly the interfacial properties of the DAC/APTES composites were never described in the manuscript. It should be also clarified in detail.

Author Response

This manuscript reports the straightforward silanization protocol for the preparation of dialdehyde cellulose hybrid composites. The experiment and their results were well discussed throughout the manuscript. However, there are lacks of the physicochemical properties related to potential applications, therefore which does not give the higher impact and also novelty on these research categories. For instance, the mechanical properties and surface nature (hydrophobic and hydrophilic) should be also kindly provided at least. In addition, importantly the interfacial properties of the DAC/APTES composites were never described in the manuscript. It should be also clarified in detail.

It was the incentive of the work (and the manuscript) to contribute to general chemical and analytical methodology in the field of renewable resources – rather than producing yet another “novel biomaterial”, as done almost ubiquitously today. We focus on the chemistry, reaction mechanisms, and the reporting of the underlying chemical processes by NMR. We – intentionally – did not focus on physicochemical / material properties, but on chemical, analytical methodology development. Physicochemical and morphological studies (hydrophilicity/hydrophobicity properties, SEM, thermostability and others) are quite extensive and will be published in a follow-up paper. An integration of these data into the present manuscript would go beyond its scope, and also a bit collide with our intention regarding its topic and purpose, so we prefer to leave the paper´s focus as it was.

Reviewer 4 Report

This manuscript reports on the direct route to silanize the dialdehyde cellulose by the silane coupling agent 3-aminopropyl-triethoxysilane in the mild aqueous condition. The experimental procedures and chemical identification have been meticulously carried out. The small question I was wondering about is what kind of morphology and properties does the silanized DAC exhibit? You used the Avicel powders to silanize and undergo freeze-drying or heat treatment drying, then has the cross-linking by the silane coupling agent changed the Avicel structure from granular to monolithic/film-like state? It would have been better if the surface energy (at least the contact angles) or the water absorbency could be reported in addition to the material appearances. Other than this point, I agree with publishing this manuscript.

Author Response

This manuscript reports on the direct route to silanize the dialdehyde cellulose by the silane coupling agent 3-aminopropyl-triethoxysilane in the mild aqueous condition. The experimental procedures and chemical identification have been meticulously carried out. The small question I was wondering about is what kind of morphology and properties does the silanized DAC exhibit?
You used the Avicel powders to silanize and undergo freeze-drying or heat treatment drying, then has the cross-linking by the silane coupling agent changed the Avicel structure from granular to monolithic/film-like state?

The topic of morphology changes and the action of periodate on the different cellulose allomorphs has been studied before. We have added an explanatory sentence to the text (first paragraph of results section) and a corresponding reference into the paper:
"Also the influence of periodate oxidation on cellulose structure and morphology, according to the cellulose allomorphs, has been studied [37].
37. Siller, M.; Amer, H.; Bacher, M.; Rosenau, T.; Potthast, A. Effects of periodate oxidation on cellulose polymorphs. Cellulose 2015, 22(4), 2245-2261."

It would have been better if the surface energy (at least the contact angles) or the water absorbency could be reported in addition to the material appearances. Other than this point, I agree with publishing this manuscript.

The final reaction product is a powder, so contact angle measurements (which would be the fastest way to check hydrophilicity/hydrophobicity) would be not feasible for this material.
To explain this, we have added one explanatory sentence (above figure 1):
"While solubilized DAC is a film-like materials after thermal drying and foam-like after lyophilization (freeze-drying), the morphology changes upon derivatization and silanized DAC is a powder."
In addition, we would like to refer to our above reply to reviewer 3. Our manuscript focused on general chemical and analytical methodology rather than on material characterization, which will be the topic of the follow-up paper.